

# Out of Asia: mitochondrial evolutionary history of the globally introduced supralittoral isopod *Ligia exotica*

Luis A. Hurtado[1], Mariana Mateos[1], Chang Wang[1,2], Carlos A. Santamaria[1,3], Jongwoo Jung[4], Valiallah Khalaji-Pirbalouty[5] and Won Kim[6]

[1] Department of Wildlife and Fisheries Sciences, Texas A&M University, College Station, TX, United States of America
[2] Department of Biology, New York University, New York City, NY, United States of America
[3] Biology Faculty, College of Science and Mathematics, University of South Florida, Sarasota, FL, United States of America
[4] Department of Science Education, Ewha Women's University, Seoul, South Korea
[5] Department of Biology, Shahrekord University, Shahrekord, Iran
[6] School of Biological Sciences, Seoul National University, Seoul, South Korea

Corresponding author
Luis A. Hurtado, lhurtado@tamu.edu

## ABSTRACT

The native ranges and invasion histories of many marine species remain elusive due to a dynamic dispersal process via marine vessels. Molecular markers can aid in identification of native ranges and elucidation of the introduction and establishment process. The supralittoral isopod *Ligia exotica* has a wide tropical and subtropical distribution, frequently found in harbors and ports around the globe. This isopod is hypothesized to have an Old World origin, from where it was unintentionally introduced to other regions via wooden ships and solid ballast. Its native range, however, remains uncertain. Recent molecular studies uncovered the presence of two highly divergent lineages of *L. exotica* in East Asia, and suggest this region is a source of nonindigenous populations. In this study, we conducted phylogenetic analyses (Maximum Likelihood and Bayesian) of a fragment of the mitochondrial 16S ribosomal (r)DNA gene using a dataset of this isopod that greatly expanded previous representation from Asia and putative nonindigenous populations around the world. For a subset of samples, sequences of 12S rDNA and NaK were also obtained and analyzed together with 16S rDNA. Our results show that *L. exotica* is comprised of several highly divergent genetic lineages, which probably represent different species. Most of the 16S rDNA genetic diversity (48 haplotypes) was detected in East and Southeast Asia. Only seven haplotypes were observed outside this region (in the Americas, Hawai'i, Africa and India), which were identical or closely related to haplotypes found in East and Southeast Asia. Phylogenetic patterns indicate the *L. exotica* clade originated and diversified in East and Southeast Asia, and only members of one of the divergent lineages have spread out of this region, recently, suggesting the potential to become invasive is phylogenetically constrained.

## INTRODUCTION

Numerous marine species have dispersed and established extensively throughout the world via marine vessels over the past several centuries (*Banks et al., 2015*; *Carlton, 1987*; *Carlton & Iverson, 1981*). The native ranges and invasion histories of a large number of them, however, remain elusive (i.e., they are cryptogenic), as a result of one or more of the following: inadequate taxonomy; poor historical documentation (particularly for older introductions); presence of cryptic lineages; and multiple inputs of invaders (*Carlton, 1996*; *Carlton, 2009*). Use of molecular data can greatly aid in the identification of their native ranges, cryptic diversity, and of the source and recipient regions (*Geller, Darling & Carlton, 2010*).

The supralittoral isopod *Ligia exotica Roux, 1828* represents a case of a widespread cryptogenic taxon with an old, albeit poorly documented, history of human-assisted dispersal (recognized as exotic in the type locality since its original description), as well as a highly problematic taxonomy. Commonly known as wharf roach, this isopod has a wide tropical and subtropical distribution, and is considered an alien species in different regions of the world, where it is frequently found in harbors, and ports, and other man-made structures (*Schmalfuss, 2003*; *Taiti et al., 2003*; *Van Name, 1936*; *Yin et al., 2013*). Similarly to the other coastal members of *Ligia*, *L. exotica* is a direct developer (i.e., lacks a planktonic larval stage; a feature of peracarids) that occupies a narrow vertical range between the supralittoral and the waterline, mainly occurring on rocky substrates (*Hurtado, Mateos & Santamaria, 2010*; *Santamaria et al., 2013*). The present-day broad distribution of *L. exotica*, including all continents except Europe and Antarctica, suggests that it possesses unique invasive capabilities within *Ligia*. With the exception of *Ligia oceanica*, an endemic of the Atlantic coast of Europe that has been introduced into some localities in the northern Atlantic coast of the US (*Richardson, 1905*), all other coastal species of *Ligia* (∼30) do not appear to have been moved by humans, or at least not to as many geographically distant places as *L. exotica* (*Schmalfuss, 2003*).

An Old World origin has been proposed for *L. exotica* (*Fofonoff et al., 2017*; *Van Name, 1936*), from where it would have been unintentionally moved around the world on wooden ships and solid ballast (*Griffiths, Robison & Mead, 2011*; *Van Name, 1936*). *Ligia exotica* was originally described by *Roux (1828)* from docks in Marseille (France), within the range of its congener *L. italica*, a species that is native and broadly distributed throughout the Mediterranean basin (*Schmalfuss, 2003*). *Roux (1828)* reasoned that a ship had likely transported this isopod from Cayenne, French Guiana (South America). Remarkably, *L. exotica* did not become established in the Mediterranean, and there are no other records of its presence in this well studied basin (*Cochard, Vilisics & Sechet, 2010*; *Fofonoff et al., 2017*; *Roman, 1977*). Roux's description places the first record of introduction of *L. exotica* at 189 years before present, but its introduction history would be older if his assertion that it was introduced from South America is correct, because this region is not regarded part of its native range. Consequently, *L. exotica* represents one of the oldest documented introductions for a marine organism. A database of 138 other coastal marine invertebrate species non-native to either Australia, New Zealand, or the United States (*Byers et al., 2015*), indicates that only two other species have older documented introduction times: the

green crab *Carcinus maenas* in 1817 (*Say, 1818*); and the hydrozoan *Cordylophora caspia* in 1799 (*Byers et al., 2015*).

*Ligia exotica* is also absent from the Atlantic coasts of Europe, where its congeneric *L. oceanica* is native and widely distributed. For this region, there is only a 1936 report of a *L. exotica* specimen found in a house in Amsterdam (*Fofonoff et al., 2017*; *Holthuis, 1949*). In addition, although a specimen assigned to *L. exotica* was collected on Sao Miguel Island (Azores) in 1905 (*Fofonoff et al., 2017*), this isopod has not become established in this archipelago, where the two European species, *L. oceanica* and *L. italica*, are present (*Cardigos et al., 2006*).

In the New World, *L. exotica* has a broad distribution along the Atlantic coast from New Jersey (US) to Montevideo (Uruguay), including the Gulf of Mexico (*Mulaik, 1960*; *Schultz, 1977*; *Schultz & Johnson, 1984*). Collections of *L. exotica* in the US Atlantic, eastern Gulf of Mexico, Brazil, and Uruguay date back to the 1880's; whereas records in the western Gulf of Mexico date back to the first half of the 20th century (*Fofonoff et al., 2017*; *Richardson, 1905*; *Van Name, 1936*). In this region, two species have been synonymized with *L. exotica*: *Ligia grandis* Perty, 1834 from Brazil; and *Ligia olfersii* Brandt, 1833 from Florida to Brazil, including the Gulf of Mexico (*Schmalfuss, 2003*). In addition, the Caribbean-endemic *Ligia baudiniana* Milne Edwards, 1840 appears to have been described based on individuals of *L. exotica* collected in Veracruz, Mexico (reviewed in *Santamaria, Mateos & Hurtado, 2014*), and the two species have been confused (i.e., *Ligia exotica* var. *hirtitarsis* Dollfus, 1890 = *L. baudiniana*; *Schmalfuss, 2003*).

Although *L. exotica* has been reported in the Pacific coast of the Americas, from the Gulf of California, Mexico, to Punta Arenas, Chile (*Van Name, 1936*), this species appears to be absent in this coast (*Fofonoff et al., 2017*). *Ligia exotica* may have been confused with *L. occidentalis*, a species native to the Gulf of California and the Eastern Pacific region between the Baja Peninsula and southern Oregon, which appears to correspond to a cryptic species complex (*Eberl et al., 2013*; *Hurtado, Mateos & Santamaria, 2010*). Despite being reported in the Gulf of California (*Mulaik, 1960*; *Richardson, 1905*), *L. exotica* was not found during a comprehensive *Ligia* collecting effort along the shores of this basin and adjacent regions (*Hurtado, Mateos & Santamaria, 2010*). *Ligia gaudichaudii* Milne Edwards, 1840, which according to its original description "seems to come from the coasts of Chile", has been synonymized with *L. exotica*, but its original locality is uncertain.

In Hawai'i, *L. exotica* was first reported in 1996, and previous records of this isopod in the archipelago correspond to *L. hawaiensis*, an endemic species (*Eldredge & Smith, 2001*). Although it may be present in other Polynesian islands (*Fofonoff et al., 2017*), the Indian and Pacific Ocean harbor a number of very similar species that have been morphologically assigned to *L. exotica*, but may correspond to different species (*Schmalfuss, 2003*; *Van Name, 1936*). In Australia, *L. exotica* is regarded as introduced in the southeastern coast, and cryptogenic in the northern coast (*Dalens, 1993*; *Fofonoff et al., 2017*; *Green, 1962*). In Africa, *L. exotica* has been reported at multiple localities. It is considered introduced into the Atlantic west-central coast and South Africa, and possibly native in the eastern coast of the continent, where it is reported from Sudan to Mozambique, including Madagascar (*Ferrara & Taiti, 1979*; *Fofonoff et al., 2017*; *Griffiths, Robison & Mead, 2011*; *Roman, 1977*).

The region spanning East Asia to the southern tip of India is also suggested to be part of the native range of *L. exotica* (*Fofonoff et al., 2017*). Molecular studies in East Asia report cryptic diversity for this isopod and propose this region as a source of introduced populations. *Jung et al. (2008)* re-assessed the previously reported (*Kwon, 1993*) occurrence of *L. exotica* in South Korea, by conducting molecular phylogenetic analyses of a fragment of the mitochondrial 16S ribosomal (r)DNA gene from individuals sampled along the South Korean coast, as well as previously reported sequences of *L. exotica* from two putative non-native populations in the US (i.e., Georgia and the Hawaiian island of O'ahu). They found two highly divergent clusters in South Korea: the "eastern group", which includes haplotypes occurring mainly along the eastern and southeastern coastlines of South Korea; and the "western group", which includes haplotypes occurring mainly along the western and southwestern coastlines of South Korea. These two lineages were in turn highly divergent from the lineage comprised of the US haplotypes. *Jung et al. (2008)* suggested that the "western group, "eastern group", and the *L. exotica* lineage from the US, each represents a distinct species, and that *L. exotica* appeared to be absent from South Korea. Their understanding on the phylogenetic relationships among the three lineages was limited, however, due to the lack of outgroups in their dataset.

*Yin et al. (2013)* conducted morphological and phylogenetic analyses of *Ligia* specimens sampled throughout the northeastern coastline of China. Their phylogenetic analyses also included the sequences examined by *Jung et al. (2008)*, and used several distant taxa as outgroups. They found two highly divergent genetic lineages, and examination of traditional morphological characters indicated that one corresponded to *L. exotica* and the other to *Ligia cinerascens* Budde-Lund, 1885. The "eastern group" sequences of South Korea, and those of Georgia and O'ahu, clustered within the *L. exotica* clade, whereas the "western group" sequences of South Korea clustered within the *L. cinerascens* clade. Within the *L. exotica* clade, two highly divergent lineages were observed, one of which contained the samples from Georgia and O'ahu, leading *Yin et al. (2013)* to suggest that East Asia was a source of introduced *L. exotica* populations.

Examination of *L. exotica* from other putative native localities, as well as from additional putative introduced populations, is needed to assess whether this isopod harbors additional molecular diversity, and to better understand its evolutionary and invasion history. An extensive dataset of *Ligia* sp. 16S rDNA sequences from Southeast to East Asia that have not been included in any published analysis is available in GenBank. Herein, we report phylogenetic analyses of these sequences, the ones reported for *L. exotica* and *L. cinerascens* from published studies, and new sequences obtained from specimens of these isopods in the Americas, Hawai'i, Africa, and Asia. Phylogenetic analyses of a subset of samples were also conducted for the mitochondrial 12S rDNA and nuclear NaK genes. We conducted phylogenetic analyses to: (1) establish whether the new sequences from Asia belong to the *L. exotica* or *L. cinerascens* clades; (2) determine whether further molecular diversity is found in these clades; and (3) shed light on the evolutionary and invasion history of *L. exotica*.

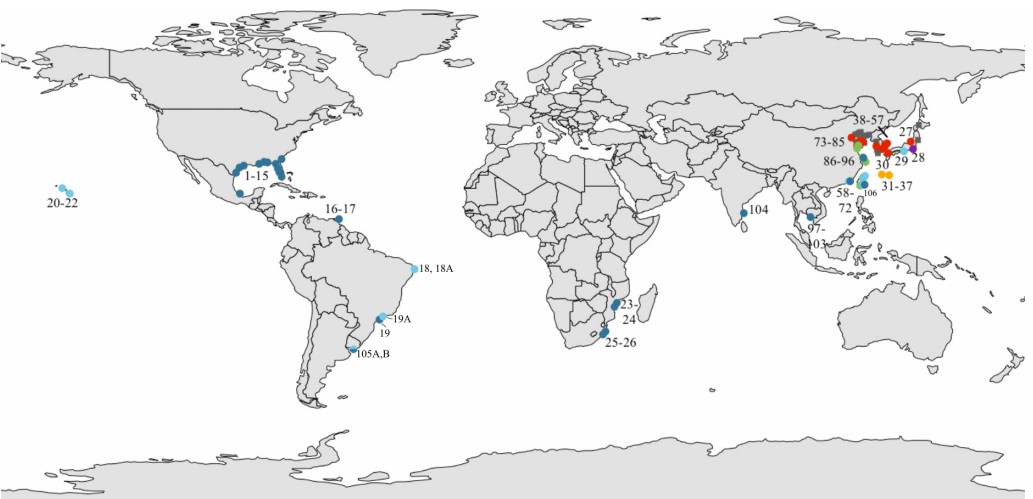

**Figure 1** **Sampled localities.** Sampled localities in (A) the global range and (B) Asia. Dots represent *L. exotica*; squares (gray) represent *L. cinerascens*. Colors correspond with lineages shown in Fig. 2. Map source: Administrative Units (admin.shp). Edition 10.1. ArcWorld Supplement, 2012. Basemap created with ArcGIS. Version 10.3, 2014; Esri, Redlands, CA, USA.

## MATERIAL AND METHODS

### Sampling

Specimens assigned to *L. exotica* were obtained from 42 localities around the world (Fig. 1; Table S1). We also obtained specimens assigned to *L. cinerascens* (from East Asia), which was used as an outgroup in the phylogenetic reconstructions. Phylogenetic analyses including most *Ligia* species (LA Hurtado, pers. comm., 2018) indicate that *L. cinerascens* is sister to the *L. exotica* clade. *Yin et al. (2013)* also found a sister relationship between *L. exotica* and *L. cinerascens*, in a dataset that also included *L. occidentalis*, and used *L. oceanica* and *Idotea baltica* (Idoteidae) as outgroups. The use of *L. cinerascens* as the only outgroup enabled the retention of a higher number of confidently-aligned characters and less homoplasy, which should enhance resolution within the *L. exotica* clade. Specimens were preserved in 70–100% ethanol. In addition to the above specimens, we used publicly available sequences (see below and in Table S1).

### DNA extraction, PCR, and sequencing

Total genomic DNA was isolated from pleopods or legs of *Ligia* specimens with the DNeasy Blood & Tissue kit (Qiagen Inc., Valencia, CA, USA) following the manufacturer's protocol. Due to its relative ease of amplification in *Ligia* and phylogenetic signal, numerous studies, including those of *L. exotica*, have reported 16S rDNA gene sequences. To maximize the number of publicly available records that could be compared, we targeted a ∼490-bp region of the 16S rDNA gene, which was amplified with published primers 16Sar (5′-CGCCTGTTTATCAAAAACAT-3′) and 16Sbr (5′-CCGGTCTGAACTCAGATCACGT-3′) (*Palumbi, 1996*). Each PCR reaction contained 1–3 µl DNA template, 0.5 µl each primer (10 pmol), 0.1 µl Taq DNA polymerase (5,000 units/µl), 0.5 µl dNTPs (10 mM), and

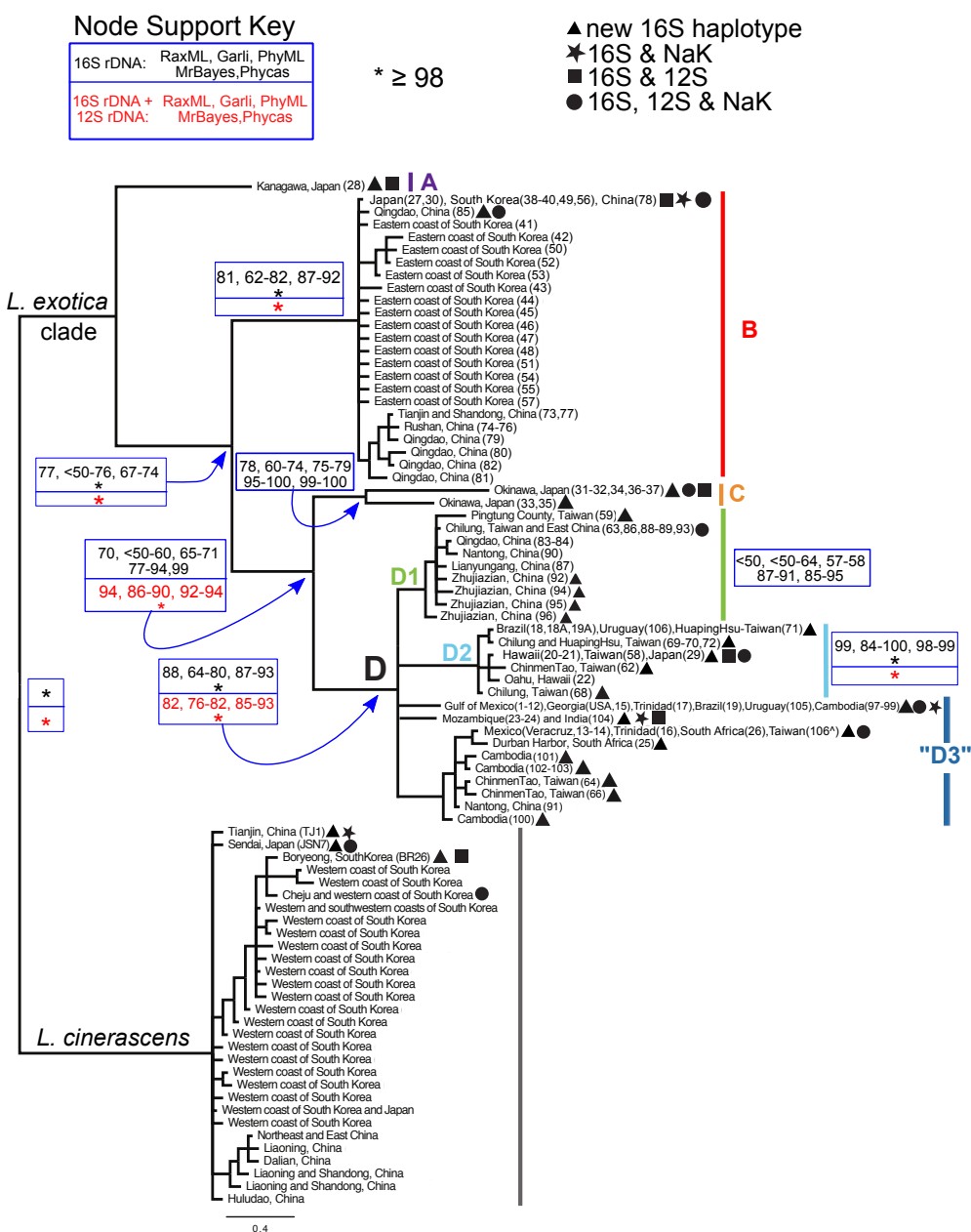

**Figure 2** **Bayesian majority consensus tree of *Ligia* samples from localities in Fig. 1.** The tree was obtained by MrBayes for 16S rDNA (model GTR + Γ), and rooted with *L. cinerascens*. Letters denote four major clades (i.e., A, B, C, and D) of *L. exotica* and three groups of haplotypes (i.e., D1, D2, and "D3") of clade D. Clade colors correspond to Figs. 1 and 3. Numbers in boxes indicate clade support value ranges for each method (bootstrap proportions and Bayesian posterior probabilities) for the 16S rDNA dataset (black font) and the 16S + 12S rDNA dataset (red font). Each range reflects pooled values obtained under different substitution models (e.g., GTR + Γ, HKY + I + Γ, and TPM2uf + I + Γ) in corresponding program. An asterisk indicates support was equal or greater than 98%. The triangles denote new haplotypes that have not been reported in the previous studies of *Jung et al. (2008)* and *Yin et al. (2013)* . Stars, squares, and circles denote 16S rDNA haplotypes for which one or more individual was examined for the 12S rDNA and/or the NaK gene. ˆindicates specimen from Taiwan for which we were only able to sequence the 12S rDNA gene.

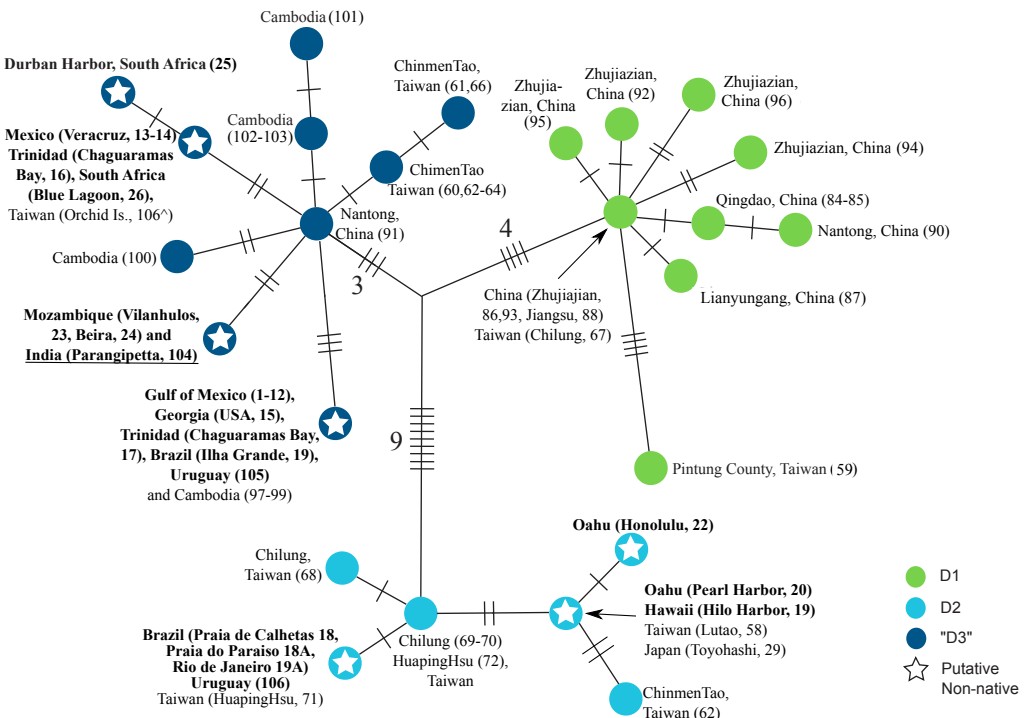

**Figure 3  Haplotype network of clade D.** Strict (unrooted) consensus of the 18 most parsimonious trees depicting the relationships among haplotypes in the clade D of *L. exotica*. Ambiguous character optimization was achieved by the accelerated transformation (ACCTRAN) algorithm. Slashes indicate the number of parsimony steps. The branch lengths within each haplogroup (i.e., D1, D2, and "D3") reflect the number of base substitutions. The numbers near the slashes correspond to the number of parsimony steps. Localities where each haplotype was found are listed next to the circles. Localities in bold are those outside the putative native range. Underlined locality label denotes uncertainty regarding its native vs. non-native status (see text). ˆindicates specimen from Taiwan for which we were only able to sequence the 12S rDNA gene (see Table S1).

2.5 µl 10 × PCR buffer (15 mM $MgCl_2$, 500 mM KCl, 100 mM Tris-HCl, pH 8.3). PCR conditions used were: 4 min at 94 °C followed by 30 cycles of 1 min at 94 °C; 30 s at 49 °C, 1.5 min at 72 °C; and a final extension at 72 °C for 4 min. PCR products were cycle sequenced at the University of Arizona Genetics Core (UAGC).

For a subset of individuals (see Table S1), we also amplified and sequenced a ∼495-bp fragment of the 12S rDNA gene (primers crust-12Sf/crust-12Sr; *Podsiadlowski & Bartolomaeus, 2005*) and a ∼709-bp fragment of the nuclear locus sodium–potassium ATPase α-subunit (NaK) (primers NaK-for-b and NaK-rev2; *Tsang et al., 2008*).

## Datasets and sequence alignment

Sequencher 4.8 (Genecodes, Ann Arbor, MI, USA) was used to assemble the new sequences and trim the primer regions. We also included all 16S rDNA sequences of *L. exotica* and *L. cinerascens* reported in *Jung et al. (2008)* and *Yin et al. (2013)*, as well as 16S rDNA sequences of specimens identified as *Ligia* sp. or *L. exotica* from Asia available in GenBank,

but not incorporated into a published study (Table S1). When present, primer regions were also removed from GenBank sequences.

All sequences were aligned in MAFFT v.7 (*Katoh, 2013*) online using the Q-INS-I strategy, which considers the secondary structure of RNA, with default parameters (e.g., gap opening penalty = 1.53). Unique haplotypes were identified on the basis of absolute pairwise distances calculated with PAUP v.4.0b10 (*Swofford, 2002*), and redundant sequences were removed from analyses. Gblocks 0.91b (*Castresana, 2000*; *Talavera & Castresana, 2007*) was used to identify positions with questionable homology that were removed prior to phylogenetic analyses. The following GBlocks parameters were used: "Minimum Number Of Sequences For A Conserved Position" = 50% of the number of sequences + 1 (i.e., 42); "Minimum Number Of Sequences For A Flank Position" = 85% of the number of sequences (i.e., 70); "Maximum Number Of Contiguous Nonconserved Positions" = 4 or 8; "Minimum Length Of A Block" = 5 or 10; and "Allowed Gap Positions" = half. In addition to the 16S rDNA only dataset, we examined a dataset of 23 taxa containing the concatenated 16S rDNA and 12S rDNA genes.

## Phylogenetic analyses

To determine the most appropriate model of DNA substitution, jModelTest v.2.1.4 (*Darriba et al., 2012*) was used to calculate likelihood scores among 88 candidate models for 16S rDNA gene, based on the fixed BIONJ-JC tree under the Akaike Information Criterion (AIC), corrected AIC (AICc), and the Bayesian Information Criterion (BIC). The best model selected by the BIC was employed in phylogenetic analyses, except in the following two cases. First, if the selected model was not available in the specific Maximum Likelihood (ML) or Bayesian Inference (BI) program, the next most complex model was implemented. Second, considering the potential problems associated with using two parameters, a proportion of invariable sites (I) and a Gamma distribution of rates among sites ($\Gamma$), simultaneously in the model (see RAxML manual and *Yang, 2006*), we chose the simpler $\Gamma$ if the best model included both I and $\Gamma$ parameters.

For the ML analyses, the CIPRES (*Miller, Pfeiffer & Schwartz, 2010*) implementations of RAxML v. 8.2.10 (*Stamatakis, 2014*) and GARLI v.2.01 (*Zwickl, 2006*) were used. RAxML executed 1,000 bootstrap replicates with a thorough ML search under the standard non-parametric bootstrap algorithm and the GTR + $\Gamma$ model, whereas GARLI implemented 1,000 bootstrap replicates, the BIC selected model, and all other settings as default. The majority-rule consensus trees for each analysis were calculated using the SumTrees command of DendroPy v.3.10.1 (*Sukumaran & Holder, 2010*). A third ML bootstrap analysis was conducted with PhyML v3.0_360 (*Guindon & Gascuel, 2003*) as implemented in a public server (http://phylogeny.lirmm.fr/phylo_cgi/one_task.cgi?task_type=phyml).

For Bayesian Inference (BI), MrBayes v.3.2.6 (*Huelsenbeck & Ronquist, 2001*; *Ronquist & Huelsenbeck, 2003*; *Ronquist et al., 2012*) as implemented in CIPRES, and Phycas v.1.2.0 (*Lewis, Holder & Holsinger, 2005a*) implemented locally, were employed. To alleviate the unpredictable behavior in Bayesian analysis when dealing with hard polytomies (i.e., "star-tree paradox"), which can lead to arbitrary resolutions and overestimation of posterior probabilities (*Alfaro & Holder, 2006*; *Kolaczkowski & Thornton, 2006*; *Lewis,*

*Holder & Holsinger, 2005b*; *Suzuki, Glazko & Nei, 2002*; *Yang & Rannala, 2005*), an analysis employing a polytomy prior was implemented in Phycas (see Phycas manual and *Lewis, Holder & Holsinger, 2005b*). The following criteria were used to determine if the Bayesian analyses had reached convergence, and if an adequate sample of the posterior had been generated: (a) the posterior probability values tended to be stable; (b) AWTY (*Nylander et al., 2008*; *Wilgenbusch, Warren & Swofford, 2004*) exhibited a high correlation between the split frequencies of independent runs; (c) the average standard deviation of the split frequencies of independent runs became stable and approached zero; (d) Potential Scale Reduction Factor (PSRF), a convergence diagnostic obtained after summarizing the sampled parameter values in MrBayes, was close to one; and (e) the Effective Sample Size (ESS) for the posterior probabilities evaluated in Tracer v.1.6 (*Rambaut et al., 2014*) exceeded 200. Samples prior to reaching stationarity were eliminated as "burn-in". The posterior probability for each node was estimated by computing a majority-rule consensus of post-burnin tree samples using the SumTrees command (*Sukumaran & Holder, 2010*).

Given the low number of alleles and shallow genetic divergences found within the clade involving haplotypes detected in putative introduced populations (see 'Results'; i.e., Clade D in Fig. 2), we also conducted a maximum parsimony branch and bound search in PAUP* v.4.0a149 (*Swofford, 2002*) for this clade. Ambiguous character optimization was achieved by the accelerated transformation (ACCTRAN) algorithm. The conservative estimate of pairwise genetic distances with Kimura-2-parameter (K2P) correction was calculated with PAUP* v.4.0a149 (*Swofford, 2002*).

## RESULTS

### Model selection

For 16S rDNA, a total of 97 sequences of the *L. exotica* clade and 41 of the *L. cinerascens* clade were examined (Table S1). The final 16S rDNA gene dataset excluding redundant sequences consisted of 81 taxa (51 in the *L. exotica* clade and 30 in the *L. cinerascens* clade). After alignment, a total of 454 characters (out of 488) were retained, for which homology was reliable, and 97 of these were parsimony informative. jModelTest selected a complex model (i.e., TPM2uf) with five substitution parameters (see jModelTest manual), +I, and +$\Gamma$ according to the AIC (weight = 0.2607) and AICc (weight = 0.3509), and a relatively simple model (i.e., HKY) with two substitution parameters (see jModelTest manual), +I, and +$\Gamma$ according to the BIC (weight = 0.3183). Similarly, the best model selected for the 16S rDNA + 12S rDNA concatenated dataset was also TPM2uf + I + $\Gamma$ (BIC weight 0.31). When applicable in the different programs used, the exact models selected by the three criteria were implemented. In addition, we implemented the GTR + $\Gamma$ model, which was included in the 99.9% cumulative weight interval of all selection criteria, in all of the methods, to assess the sensitivity of clade support values to variations in the substitution model (Table S2).

### Phylogenetic results

In general, the use of different substitution models or priors yielded similar overall topologies of phylogenetic trees, although some discrepancies, reflected in node support
**Table 1  Genetic divergences among major lineages within *L. exotica* and *L. cinerascens*.** Conservative estimates of evolutionary divergence among major lineages within *L. exotica* and *L. cinerascens*, as measured by percent Kimura-2-parameter distances. Lower matrix: distance range. Upper matrix: average distance. Values on diagonal show minimum and maximum within-clade divergence. Empty cells: no ranges available because selected clade was represented by a single sample.

| | *L. exotica* clade A | *L. exotica* clade B | *L. exotica* clade C | *L. exotica* clade D | *L. cinerascens* |
|---|---|---|---|---|---|
| *L. exotica* clade A | – | 11.5 | 12.5 | 10.5 | 10.4 |
| *L. exotica* clade B | 11.1–12.1 | **0.2–2.0** | 8.8 | 10.0 | 11.7 |
| *L. exotica* clade C | 11.9–13.2 | 7.3–10.8 | **6.3** | 7.6 | 13.6 |
| *L. exotica* clade D | 9.8–11.1 | 8.3–11.6 | 6.7–9.2 | **0.2–4.6** | 13.0 |
| *L. cinerascens* | 9.4–11.0 | 10.8–13.1 | 12.3–15.0 | 11.6–14.9 | **0.1–2.9** |

values (Fig. 2; Table S3), were observed among different approaches. Our phylogenetic reconstructions (Fig. 2) recovered a highly supported split (Bootstrap Support (BS): 98–100; Posterior Probability (PP): 100) between *L. exotica* and *L. cinerascens*. The *L. cinerascens* clade is restricted to the northern part of East Asia, in the western coast of South Korea, Honshu and Hokkaido in Japan, and northeastern China. Maximum K2P divergence observed within this clade was 2.9% (Table 1). The NaK gene was obtained for 20 individuals representing most of the main lineages of the *L. exotica* clade (see Table S1; Fig. 2), as well several individuals assigned to *L. cinerascens*. Three fixed differences were detected between the *L. exotica* clade and *L. cinerascens*, but no variation within them was found.

Our analyses revealed 23 new 16S rDNA haplotypes within the *L. exotica* clade (marked with triangles in Fig. 2) that were not reported in the previous studies of *Jung et al. (2008)* and *Yin et al. (2013)*. The *L. exotica* clade was divided into four main lineages (named A, B, C, and D). Node support for different datasets (i.e., 16S rDNA alone and 16S rDNA + 12S rDNA), methods and substitution models is shown in Table S3, and summarized in Fig. 2. In general, the main clades (B, C, and D) received high support from all analyses except the ML analyses of the 16S rDNA dataset alone (see Fig. 2). Divergences between and within main lineages are shown in Table 1. At the base of the *L. exotica* clade, a relatively distant (K2P divergence = 9.8–13.2%) lineage from Kanagawa, Japan (A) diverged from a clade that contains the remaining lineages (clade B + C + D; high support from all analyses except ML of 16S rDNA). Within the latter clade, a basal split (K2P divergence = 7.3–11.6%) is observed between a lineage consisting mainly of samples from temperate regions in East Asia (clade B; maximum within-clade K2P divergence = 2.0%) and a clade (i.e., C + D) containing the remaining lineages. Some of the populations in Clade B have overlapping distributions with *L. cinerascens* in China (e.g., Tianjin and Shandong) and the western coastline of South Korea (e.g., Boryeong) (Fig. 1). Within the clade C + D, a basal divergence (K2P = 6.7–9.2%) is observed between a lineage from Okinawa, Japan (C), which contains two highly divergent lineages from this island (6.3% K2P divergence), and a clade (D) with the remaining samples (maximum within-clade K2P divergence = 4.6%). Within clade D, several lineages are distinguished. The first (D1 in tree) is restricted to East Asia localities (maximum within-clade K2P divergence = 1.3%; support from ML was weak). The second (D2) has haplotypes found in East Asia, but also in introduced populations from Hawai'i, Brazil, and Uruguay (maximum within-clade K2P

divergence = 0.9%; well supported by all methods). The remaining haplotypes formed a clade with a subset of the methods, but support was weak. We have therefore collapsed it in Fig. 2, but labeled all these haplotypes as belonging to haplogroup "D3" (maximum within-haplogroup K2P divergence = 1.1%). Haplogroup "D3" has haplotypes observed in putative introduced populations from the Gulf of Mexico, Trinidad, Brazil, Uruguay, South Africa, Mozambique, and is also found in South to East Asia (see 'Discussion' for considerations of native range and introduced populations).

Figure 3 shows a strict consensus unrooted parsimony tree (made of the 18 most parsimonious trees; CI excluding uninformative characters = 0.8421; RI = 0.9552) for clade D (i.e., the only clade found to contain haplotypes found in putative introduced populations). The three previously described main lineages within this clade are represented by different haplotype colors (i.e., D1 green circles, D2 light blue circles, and "D3" dark blue circles). Seven haplotypes were observed in putative introduced populations (see 'Discussion'), three within D2 and four within "D3" (denoted by stars). "D3" contains the haplotype that was most common in introduced populations of the Gulf of Mexico, and was also found in the US Atlantic coast (Georgia), Trinidad (Chaguaramas Bay), Brazil (Ilha Grande, Rio de Janeiro), Uruguay, and Cambodia. Another D3 haplotype was found in Mexico (Veracruz), Trinidad (Chaguaramas Bay), and South Africa, but was not observed in Asia. A third haplotype was observed in Mozambique, which likely represents another introduced population, and in India. The fourth putatively introduced D3 haplotype was only observed in South Africa. Within D2, a haplotype was found in O'ahu (Pearl Harbor) and Hawai'i Island, which was also observed in Japan and Taiwan. Another D2 haplotype was found exclusively in O'ahu (Honolulu Harbor). Finally, a third D2 haplotype was observed in Brazil (Praia de Calhetas, Cabo de Santo Agostinho, Pernambuco), Uruguay, as well as in Taiwan.

## DISCUSSION

### Multiple divergent lineages and taxonomic uncertainty

The *L. exotica* clade is comprised of highly divergent lineages, which probably represent multiple species. Using morphological characters (i.e., number of segments in the second antenna flagellum, uropod, characters of the telson and the shape of the appendix masculina on the second pleopod of adult males), *Yin et al. (2013)* concluded that members of clades B and D in our phylogenetic tree correspond to *L. exotica* (they did not examine members of clades A and C). Thus, it is possible that cryptic diversity occurs within the *L. exotica* clade. High levels of cryptic diversity have been reported in numerous studies of *Ligia* and other intertidal isopods regarded as single broadly distributed species (*Hurtado, Lee & Mateos, 2013*; *Hurtado, Mateos & Liu, 2017*; *Hurtado et al., 2016*; *Hurtado, Mateos & Santamaria, 2010*; *Santamaria et al., 2017b*; *Santamaria et al., 2016*; *Santamaria, Mateos & Hurtado, 2014*; *Santamaria et al., 2013*).

Some of the lineages within the *L. exotica* clade, however, may correspond to species that have been described in the East Asia region. For example, our Clade C samples, from Okinawa and Kitadaito, may correspond to *Ligia ryukyuensis Nunomura, 1983*,

described from the Ryukyu Islands (*Nunomura, 1983*), and/or *Ligia daitoensis Nunomura, 2009*, described from the Daito Islands (*Nunomura, 2009*). Similarly, our sample from Kanagawa (Clade A) may correspond to *Ligia yamanishii Nunomura, 1990* described from the Tokyo Prefecture (*Nunomura, 1990*). South of Kaganawa, *Ligia miyakensis Nunomura, 1999* and *Ligia hachijoensis Nunomura, 1999* are also reported, both described from the Izu Islands (*Nunomura, 1999*); and *Ligia boninensis Nunomura, 1979*, described from the Bonin Islands (*Nunomura, 1979*), south of the Izu Islands. *Schmalfuss (2003)* indicates, however, that the description of *L. miyakensis* does not allow separation from *L. exotica*, and that *L. hachijoensis* is possibly conspecific with *L. exotica*. Unfortunately, the condition of our specimens precluded adequate examination of their morphology, and future work is needed to determine whether some of our lineages represent these species. Given the taxonomic uncertainty, and to facilitate the discussion of our results, however, we refer to lineages A, B, C, and D collectively as the *L. exotica* clade.

## Native range and introduced populations

The observed phylogenetic patterns support an origin and long evolutionary history of the *L. exotica* clade in the East and Southeast Asia region. Its sister relationship with *L. cinerascens,* also distributed in East Asia, suggests that their ancestor occupied, and diversified within, this region. Furthermore, a long evolutionary history of the *L. exotica* clade within this region is also supported by the numerous diversification events that led to highly divergent lineages, all of which, except for seven haplotypes within clade D, are only found in this region. Clade D exhibits much higher genetic diversity within the East and Southeast Asia region than in all other sampled regions collectively (i.e., the Americas, Hawai'i, Africa and India), where only seven out of the 25 16S rDNA haplotypes found in clade D were detected. Three of these seven haplotypes were also observed in East and Southeast Asia. The other four, albeit not detected in this region, were only separated by few substitutions (1–3 mutational steps away) from haplotypes found in East and Southeast Asia, and it is possible that we failed to sample them in this region (individuals from Veracruz, which had one of these four haplotypes have also the same 12S rDNA haplotype found in an individual from Taiwan). Therefore, our results suggest the *L. exotica* clade originated and diversified in East and Southeast Asia, and that recently, relative to the diversification observed in this clade, members of Clade D have spread out of this region.

Although South Asia and the eastern coast of Africa have been suggested to be part of the native range of *L. exotica* (*Fofonoff et al., 2017*), it is likely that the *L. exotica* populations distributed there are introduced. Only one 16S rDNA haplotype was observed in these two regions, which was not found in East and Southeast Asia, but is only separated by two nucleotide differences from one observed in China. Finding the same haplotype between these two distant regions (the distance between the localities in Mozambique and India is ~6,000 Km) suggests that the specimens from Mozambique, at least, are non-native. South Asia and the eastern coast of Africa harbor endemic species or lineages of other *Ligia* species, and species in the Indian Ocean have often been misidentified as *L. exotica* (*Schmalfuss, 2003*; *Taiti, 2014*). *Ligia exotica*, thus, may not be as common as previously thought in these regions, and scattered isolated introduced populations might occur within the range

of native lineages, as observed in the Caribbean (see below). South Asia is home to *Ligia dentipes* Budde-Lund, 1885, which has a broad distribution that spans the Nicobar Islands, Andaman Islands, Maldives, Seychelles, Sri Lanka, and Thailand (*Santamaria et al., 2017b*; *Taiti, 2014*). Three divergent (12–15% divergence at the COI gene) lineages of *L. dentipes* were detected in a study that surveyed the Seychelles, Sri Lanka, and Thailand (*Santamaria et al., 2017b*). Similarly, the eastern coast of Africa harbors two highly divergent lineages of *Ligia vitiensis* (Dana, 1853), one distributed in Tanzania, Seychelles, and Madagascar, and the other in Tanzania (*Santamaria et al., 2017b*). Other species reported in East Africa, but lacking molecular data, are: *Ligia ferrarai* Kersmaekers & Verstraeten, 1990 in Madagascar; *Ligia pigmentata* Jackson, 1922 in Somalia (also reported in the Red Sea and Persian Gulf; although records for this last basin have been questioned; *Khalaji-Pirbalouty & Wägele, 2010*); and *Ligia malleata* Pfeffer, 1889 in Tanzania, which is possibly a synonym of *L. exotica* (*Schmalfuss, 2003*).

*Ligia exotica* is considered introduced in South Africa (*Griffiths, Robison & Mead, 2011*), where we found two haplotypes, differing at a single nucleotide position from each other, belonging to haplogroup "D3". One of these haplotypes was also observed in Mexico and Trinidad. Three species of *Ligia* are native to South Africa: *Ligia dilatata* Brandt, 1833 (also reported in Namibia); *Ligia glabrata* Brandt, 1833 (also reported in Namibia); and *Ligia natalensis* Collinge, 1920 (*Schmalfuss, 2003*). These species appear to have a long evolutionary history in South Africa (*Greenan, Griffiths & Santamaria, 2017*). *Ligia exotica* populations in the Atlantic west-central coast of Africa are also considered introduced, although genetic studies would be useful to verify species identity (*Fofonoff et al., 2017*). *Ligia exotica* also does not appear to be native in Southwest Asia, and there is doubt about reports of this isopod in the Red Sea (*Khalaji-Pirbalouty & Wägele, 2010*). The region has several endemic *Ligia* species reported: *Ligia dioscorides* Taiti & Ferrara, 2004 from the Socotra Archipelago in Yemen; *Ligia persica* Khalaji-Pirbalouty & Wägele, 2010 from the Persian Gulf; and *Ligia yemenica* Khalaji-Pirbalouty & Wägele, 2010 from the Gulf of Aden (*Khalaji-Pirbalouty & Wägele, 2010*).

Pacific populations outside East and South East Asia are also likely introduced. One of the two *L. exotica* haplotypes found in Hawaii was also observed in East Asia (Taiwan and Japan), and the other one differs at a single nucleotide position. As in the Indian Ocean, a number of different species in the Pacific Ocean may have been wrongly assigned to *L. exotica* (*Schmalfuss, 2003*; *Van Name, 1936*). Although we did not examine individuals from Australia, it is likely that populations of *L. exotica* in this continent are also introduced. Two endemic species are reported there: *Ligia australiensis* Dana, 1853, which is widely distributed in the coast of Australia, including Tasmania and Lord Howe Island; and *Ligia latissima* (Verhoeff, 1926), endemic to New Caledonia (*Schmalfuss, 2003*). Future work is needed to genetically characterize native and non-native *Ligia* from Australia. Interestingly, despite reports of the occurrence of *L. exotica* in the Gulf of California (*Mulaik, 1960*; *Richardson, 1905*), we failed to find it during extensive surveys of this and the adjacent regions (*Eberl et al., 2013*; *Hurtado, Mateos & Santamaria, 2010*). Although it is possible that *L. exotica* occurs in hitherto unsampled Pacific coast localities of the

New World, it is likely that past records of this species were misidentifications of the morphologically similar species *L. occidentalis.*

In the Americas, *Ligia exotica* is very common in the US Atlantic coast, Gulf of Mexico, and the coastal region between Brazil and Argentina, where other *Ligia* species are rare or absent. Records of *L. exotica* in the US Atlantic, eastern Gulf of Mexico, Brazil and Uruguay date back to the 1880's, and in the western Gulf of Mexico to the first half of the 20th century (*Fofonoff et al., 2017*; *Richardson, 1905*; *Van Name, 1936*). Within the Gulf of Mexico (a mostly sandy coastline), jetties and other man-made structures have provided suitable habitats for this isopod throughout the basin (*Schultz & Johnson, 1984*). Most of this basin is devoid of other *Ligia* species, with the exception of a few localities in Florida and Yucatán, where *L. baudiniana* is present (*Santamaria et al., 2017a*; *Santamaria, Mateos & Hurtado, 2014*; LA Hurtado, pers. comm., 2018). *Ligia exotica* exhibits very low genetic diversity in this region, with a single 16S rDNA haplotype observed, except for Veracruz, where a different closely related haplotype was detected (both from the "D3" haplogroup). The most common haplotype was also observed in Georgia, in the Atlantic coast of the US, where *L. exotica* is also broadly distributed from New Jersey to Florida in the absence of other *Ligia*, with the exception of the southern tip of Florida where *L. baudiniana* is also reported (*Schultz & Johnson, 1984*).

In the Caribbean, we found *L. exotica* only in a small pile of rocks in a little harbor in Trinidad, despite a major sampling effort for *Ligia* that included different countries in the region, where the widely distributed native *L. baudiniana* was mainly recovered (*Santamaria, Mateos & Hurtado, 2014*). Two haplotypes were found in Trinidad, one was also observed in Veracruz, Mexico, and South Africa, whereas the other was also observed in the Atlantic US, Gulf of Mexico, Brazil, Uruguay, and Cambodia. It is possible that some of the previous reports of *L. exotica* in the Caribbean correspond to misidentifications, as this species has been confused with *L. baudiniana* (*Santamaria, Mateos & Hurtado, 2014*; *Schmalfuss, 2003*; *Van Name, 1936*).

In the Atlantic coast between Brazil and Argentina *L. exotica* appears to be broadly distributed (*Schmalfuss, 2003*) in the absence of native *Ligia* (although *L. baudiniana* has been reported in Rio de Janeiro (*Van Name, 1936*), this needs to be confirmed; we only found *L. exotica* at this and a nearby locality). We sampled five localities in this region and found one haplotype from clade D2 (also found in Taiwan) and one from haplogroup "D3" (identical to the most common haplotype found in the Gulf of Mexico). The presence of two divergent haplotypes (separated by 16 nucleotide differences at the 16S rDNA gene) suggests independent introductions have occurred in this region. Both haplotypes can co-occur in close sympatry. In Uruguay, the two haplotypes were observed in specimens collected concurrently from the same rock.

## Phylogeographical patterns in East and Southeast Asia

Occurrence of multiple genetically divergent lineages within the *L. exotica* clade in East and South East Asia is similar to the phylogeographic patterns observed in the following recognized species of *Ligia*, whose distribution includes or is limited to tropical and/or subtropical coasts of other regions: *L. occidentalis*, whose range spans the Pacific coast

between central Mexico and southern Oregon, including the Gulf of California (*Eberl et al., 2013*; *Hurtado, Mateos & Santamaria, 2010*); *L. baudiniana* in the Caribbean and a small Pacific region between Central and South America (*Santamaria, Mateos & Hurtado, 2014*); *L. hawaiensis* in the Hawaiian archipelago (*Santamaria et al., 2013*); and *L. italica* in the Mediterranean basin (LA Hurtado, pers. comm., 2018). The relatively high genetic diversity of the *L. exotica* clade contrasts with the low diversity observed in its sister lineage *L. cinerascens* (maximum K2P divergence within this species = 2.9%), suggesting different evolutionary histories. One evident difference between the two lineages is their geographic distributions. Within our study area alone, *L. cinerascens* was generally found in relatively colder (mostly temperate) regions, including the northern Yellow Sea, Bohai Sea, Korean Peninsula, and the northern portion of the Japanese archipelago. The range of *L. cinerascens* extends further north into the Kuril Islands (*Yin et al., 2013*) and the Peter de Great Gulf (i.e., the southernmost part of Russia in the Sea of Japan; *Zenkevich, 1963*). Although the ranges of *L. exotica* and *L. cinerascens* overlap (Fig. 1), *L. exotica* is generally found in warmer (tropical and subtropical) regions. Due to its distribution at higher latitudes, the lower genetic diversity of *L. cinerascens* may reflect a history of recent extinction-expansion events associated with glacial and postglacial cycles. A similar pattern of recognized species of *Ligia* from high latitudes (at least in the northern hemisphere) harboring low genetic diversity occurs in *L. pallasi* (*Eberl, 2013*) and *L. oceanica* (*Raupach et al., 2014*).

Within the *L. exotica* clade, Clade B, which is mostly restricted to temperate areas, exhibits comparatively lower genetic diversity (maximum K2P divergence = 2.0) than clades C and D, which occur in warmer regions. Lineage A was found only in Kanagawa, Japan. The pattern of comparatively lower diversity within Clade B, whose distribution overlaps with part of the range of *L. cinerascens*, may also be explained by a history of recent extinction-recolonization events associated with glacial cycles. A similar pattern of reduced genetic diversity at higher latitudes within a recognized coastal isopod species occurs in the northernmost clade of *L. occidentalis* in California (*Eberl et al., 2013*), as well as in the northernmost clade of the supralittoral isopod *Tylos punctatus*, between Southern California and the Baja Peninsula (*Hurtado et al., 2014*).

Temperature also appears to be an important factor determining the distribution of the other *L. exotica* lineages, which are found in warmer waters. Although the northern distribution of *L. exotica* Clade D1 overlaps with the southern range of Clade B in the Yellow Sea, Clade D1 was detected as far south as Taiwan. Clade D2 was found in warmer waters. A haplotype of this clade was observed in the southern coast of Honshu, Japan, which is in a region with warmer water, and was also found in Taiwan and Hawai'i. The only locality where lineage A was found is also in the southern coast of Honshu. Haplogroup "D3" was restricted to warmer waters and reached the southernmost areas (i.e., Cambodia) in what appears to be the native range of the *L. exotica* clade. Sea surface temperature (SST) appears to be an important factor determining the distribution of lineages in *L. occidentalis*. In this isopod, the geographical limit between two main clades largely reflects the changes in SST that define the Point Conception biogeographical boundary in California (*Eberl et al., 2013*). Although coastal *Ligia* are essentially terrestrial and do not venture into open water,

SST influences abiotic factors likely important to their survival and reproduction, such as air temperature, sea and land breezes, atmospheric humidity and coastal fog (*Eberl et al., 2013*).

A dynamic past geological history in the Southeast-East Asia region (*Ni et al., 2014*; *Wang, 1999*) may have contributed to divergences within the *L. exotica* clade, but we cannot pinpoint specific events. Opportunities for long-standing isolation and differentiation appear to have occurred in the Japanese archipelago, as suggested by the divergent lineages found in our analyses, and by the reports of several endemic *Ligia* species to this region (*Nunomura, 1979*; *Nunomura, 1983*; *Nunomura, 1999*), discussed above. The highly complex geological history of the Japanese archipelago is considered crucial in the generation and maintenance of the high species diversity and endemism of this region (reviewed in *Tojo et al., 2017*), considered a global hotspot of biodiversity (*Ceballos & Brown, 1995*; *Conservation International, 2016*). Such history has been associated with the presence of multiple highly divergent lineages in the also supralittoral isopod *Tylos granulatus* (*Niikura, Honda & Yahata, 2015*), the sandy beach amphipod *Haustorioides japonicas* (*Takada et al., 2018*), as well as in multiple insects (*Tojo et al., 2017*). It is important to conduct a thorough examination of *Ligia* in the Japanese archipelago, which likely will reveal additional diversity and will help to establish the distribution limits of divergent lineages that appear to be endemic to this region (i.e., A and C). Relatively deeper divergences within Clade D also suggest greater opportunities for diversification have occurred in the warmer waters. The island of Taiwan also exhibits high levels of genetic diversity, with the presence of multiple divergent lineages, as observed in the present study and in a previous study based on the Cytochrome Oxidase I (COI) gene (*Chang, 2013*).

## Evolution of 'invasiveness'

Haplotypes found at putative introduced populations are restricted to clade D, and within this clade, to haplogroups D2 and "D3". Therefore, the potential to become invasive appears to be phylogenetically constrained, and to have arisen recently relative to the diversification observed in the *L. exotica* clade. A similar pattern is observed in the leafmining global fly pest *Liriomyza sativae*, in which all invasive populations fall within a single clade (*Scheffer & Lewis, 2005*).

The inherent traits that may enable certain genetic backgrounds of *L. exotica* to become established at a non-native location might include higher tolerance to environmental stresses associated with the journey and/or the new locality. Tolerance of higher environmental temperatures (at least compared to *L. cinerascens* and *L. exotica* clades A and B) might be associated with successful dispersal and establishment. Essentially, all the introduced populations of *L. exotica* are found in tropical to subtropical locations. Environmental similarity between donor and recipient regions might increase the chance of a successful invasion (*Seebens, Gastner & Blasius, 2013*). Nonetheless, lineages of *L. exotica* distributed in similarly warm waters (i.e., C and D1) are not found in introduced populations. Their absence could simply reflect a lack of opportunity to "hitch a ride". This might be a reasonable explanation for clade C, as it is only known from Okinawa, but D1 has a relatively broader distribution in East Asia, that overlaps with that of D2 and "D3".

Tolerance to desiccation might also be associated with invasive ability in *L. exotica*. *L. exotica* individuals were likely unintentionally loaded onto ships along with ballast stones commonly used during the 18th and 19th centuries, and dumped at the destination port (*Griffiths, Robison & Mead, 2011*; *Van Name, 1936*). Isopods riding in the holds of ships likely faced limited access to seawater. Low desiccation resistance is a feature of the genus *Ligia*, constituting one of the factors that constrain its coastal distribution to a very narrow vertical range between the supralittoral and the water line (*Carefoot & Taylor, 1995*; *Hurtado, Mateos & Santamaria, 2010*). A superior desiccation resistance and osmoregulation ability compared to *L. taiwanensis* and/or *L. cinerascens*, which could enhance survival of such journeys, has been reported in *L. exotica* from Taiwan (*Tsai, Dai & Chen, 1997*; *Tsai, Dai & Chen, 1998*), where clade D occurs. Once in a new harbor, the availability of rocky habitat, similar temperatures to source localities, and high reproductive rates would have contributed to their successful establishment. Indeed, high reproductive rates have been reported for *L. exotica* in an introduced Brazilian population (*Lopes et al., 2006*).

Finally, *L. exotica* do not appear to have evolved traits that enable them to outcompete and displace native *Ligia* species. In some regions where other *Ligia* species are widely distributed, establishment of introduced *L. exotica* populations has failed (e.g., the Mediterranean, Atlantic Europe, the Azores), or only few scattered introduced *L. exotica* populations have established, mainly in man-made rocky habitats (e.g., Hawaii and the Caribbean). It is possible that the broad distribution of endemic *L. occidentalis* lineages in the Gulf of California and Pacific coast between central US and southern Mexico precludes the establishment of *L. exotica* in these regions. In contrast, absence of other *Ligia* species may have favored the establishment and wide expansion of *L. exotica* in the US Atlantic coast, the Gulf of Mexico, and the coast between Brazil to northern Argentina.

## CONCLUSION

The present study capitalized on a large dataset of 16S rDNA sequences for *Ligia* specimens from East and Southeast Asia. Addition of *de novo* sequences from other localities within this region and putative introduced populations around the world, allowed for a broad geographic representation of the widespread *L. exotica*. Phylogenetic analyses revealed that the *L. exotica* clade originated and diversified in East and Southeast Asia, and only members of one of the divergent lineages have spread out of this region recently, suggesting that the potential to become invasive is phylogenetically constrained. Much higher haplotype diversity was observed in East and Southeast Asia, than in the other regions surveyed (Americas, Hawai'i, Africa and India), where only seven 16S rDNA haplotypes were detected; which were identical or very closely related to haplotypes from East and Southeast Asia. Multiple geographically distant introduced populations share the same mitochondrial haplotype, but in the New World at least three haplotypes arrived. This study also revealed interesting biogeographical patterns, such as the reduced genetic diversity at higher latitudes. Our study demonstrates the potential of even modest genetic information collected at broad scales, to substantially improve our understanding on the evolutionary and invasive histories of cryptogenic species.

## ACKNOWLEDGEMENTS

We thank the following individuals and institutions for specimens: Aska Yamaki (Yokohama National University, Japan); Jeng-Di Lee (Institute of Marine Affairs, National Sun Yat-sen University, Kaohsiung, Taiwan); Dr. Ravichandran (Annamalai University, India), Charles Griffiths; Jesser Fidelis de Souza Filho (Museu de Oceanografia/UFPE, Recife, PE, Brazil); Paulo C. Paiva and Gustavo Mattos (Departamento de Zoologia, Universidade Federal do Rio de Janeiro, Rio de Janeiro, Brazil); I Tomasco; D Macedo; Florida Museum of Natural History.

### Funding

Funding was provided by an NSF grant DEB 0743782 to Luis A. Hurtado and Mariana Mateos, TAMU-CONACyT grants to Luis A. Hurtado, and Hispanic Leaders in Agriculture and the Environment Program Fellowship and Community Impact Funds to Carlos A. Santamaria. The funders had no role in study design, data collection and analysis, decision to publish, or preparation of the manuscript.

### Grant Disclosures

The following grant information was disclosed by the authors:
NSF: DEB 0743782.
TAMU-CONACyT.

### Competing Interests

The authors declare there are no competing interests.

### Author Contributions

- Luis A. Hurtado and Mariana Mateos conceived and designed the experiments, performed the experiments, analyzed the data, contributed reagents/materials/analysis tools, prepared figures and/or tables.
- Chang Wang performed the experiments, analyzed the data, contributed reagents/materials/analysis tools, prepared figures and/or tables, authored or reviewed drafts of the paper.
- Carlos A. Santamaria, Jongwoo Jung and Valiallah Khalaji-Pirbalouty performed the experiments, contributed reagents/materials/analysis tools, authored or reviewed drafts of the paper.
- Won Kim contributed reagents/materials/analysis tools, authored or reviewed drafts of the paper.

### DNA Deposition

The following information was supplied regarding the deposition of DNA sequences:
The new sequences described here are accessible via GenBank accession numbers KX447715–KX447756 and MG676400–MG676443. Details are provided in Table S1.

## Data Availability

The research in this article did not generate any code. The raw data (i.e., DNA sequences) have been deposited in GenBank. The DNA sequence alignment is available in the Supplemental Materials as Dataset S1.

## Supplemental Information

Supplemental information for this article can be found online at http://dx.doi.org/10.7717/peerj.4337#supplemental-information.

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
