# Peer review of "Out of Asia: mitochondrial evolutionary history of the globally introduced supralittoral isopod Ligia exotica"

_PeerJ, doi:10.7717/peerj.4337_

## Round 0.1 · original submission · Major Revisions

Reviewer #1 is of the opinion that studies based on just one short mtDNA fragment are not publishable these days, as the findings are unlikely to be robust. If there is a way to get additional sequence information (even for a subset of the samples), that could provide more compelling evidence in support (or not) of the hypotheses. If the authors must rely on only the one sequence, I think limitations need to be addressed early in the manuscript (in the Introduction); preferably, some data from another gene can be added. I agree with reviewer # 2 that the absence of samples from Europe leads to a major data gap. I think if the authors could add sequences from Europe, conclusions about the native range could become substantially stronger. Accommodating these reviewer requests could potentially change some conclusions, so I would consider the requested revisions to be major.

Additional specific comments

L 30 “highly genetically divergent lineages” – I suggest “highly divergent genetic lineages”
L 36 “nonnative populations” – suggest “putative nonnative populations”
L 37 “suggesting that the potential to become invasive is phylogenetically constrained” – I don’t see a strong basis for this statement. Given that your putative exotic samples come mainly from warmer regions, if haplotypes are indeed associated with temperature (as proposed in the Discussion), then the absence of sampling from cooler parts of the putative exotic range would miss the invasvie cool-adapted haplotypes. This issue also affects the argument presented on lines 451-453.
L 56 Can you briefly explain the basis for this proposal?
L 117 same as L 30
L 154 “as outgroup” change to “as an outgroup”
L 477-479 I wonder if use of adjectives such as “putative” or “presumed” should be used when referring to invasive and native?
L 482 remove “dramatically”
Fig 2 – several new specimens are labeled “east Asia”. Can more specific localities be provided for these?

Reviewer 1 ·

Basic reporting

This MS uses professional English, provides a very good literature coverage, has a good structure, and offers the necessary raw data in the supplement.
Nevertheless, it fails in one crucial issue: the results obtained by a short mtDNA gene fragment cannot actually test authors' hypotheses in a suficient way, so that all further discussion is redundant.

Experimental design

Experimental design and methods, if we exclude the number and length of gene fragments used, are very good, using state-of-the-art phylogenetic approaches and tools. The research question is valid and interesting, but again, the problem lies with the number of genes targeted, which undermines the whole endeavor.

Validity of the findings

As arleady mentioned, the main problem with this MS is the data used for testing hypotheses. Today, taking into account the availability and reduced cost of relevant analyses, sequences of such a short mtDNA fragment cannot be considered as adequate evidence to test hypotheses of phylogeography and origins of clades. Therefore, I suggest that the authors continue with several more genes, including nuclear ones, or even applying RADseq approaches, so that they are able to support their claims unambiguously.

Additional comments

Given the availability of samples from various localities, I believe that the authors can add more gene markers to their data quite easily and soon, so that their work gains more power and validity. Of course, I see that such data may not be available for all localities (for which GenBank sequences were used here), but I'm sure it's relatively easy to obtain fresh Ligia material from at least some of these via colleagues or even amateurs at the respective regions, given that these animals are easy to sample. I'm sure that even if the authors do not manage to include all localities for which 16S data are available online, the inclusion of more gene markers and, maybe, novel NGS approaches will render their work far more powerful and robust.
A less important issue is that of terminology used for L. exotica, which is characterised as 'invasive' in many parts of the text. The authors might consider using another term, like 'exotic' or 'alien', given that invasiveness has not been actually documented for this species, and in fact, it has been assigned this label only in eastern USA coasts (again without much documentation). Invasive species should have been shown to negatively affect local biodiversity, ecosystems or human economy. If this has not been documented or does not happen, then the species are just 'alien' or 'exotic'.

Reviewer 2 ·

Basic reporting

This is a very interesting dataset from a very successful invasive species. However it seems that the European distribution of the species was completely ignored throughout the manuscript and the only mention to this in Introduction hypothesizes that it has originated in the region. The authors should be more cautious in their discussion since their study area does not include any European locality. The idea mentioned in lines 128-132 is repeated in lines 314-317 and supported only with self citations (6-7 studies). Technically there is nothing wrong however in my opinion this should be avoided.

Experimental design

no comment

Validity of the findings

Although based on the data available the native range of L. exotica seems to be located in Asia, the entire European distribution of the species is not included in this study and there should be a reference to this in Discussion. Specially since before the closure of the Tethys Sea these two regions were connected. I think, based on the data, the most objective conclusion would be that the origin of the species is not in the new world. The fact that unpublished phylogenetic analyses retrieved L. exotica and L. cinerascencis (only distributed in Asia) as sister taxa does not in itself support an Asian origin. First there are many species in Ligia genus and it is not clear whether they were all included in the mentioned (unpublished) phylogenetic reconstruction, second if these results are based in only one mitochondrial gene they could also be meaningless. For this reason I think the phrase "Thus, the reciprocal monophyly of L. cinerascens and L. exotica is well-supported" should be removed. However most of the species of Ligia seem to be restricted to Asia (based on WORMS) which suggests that most of the genus diversified in the region and the authors could use this argument in their favour. Discussion between lines 376 and 414 is also somewhat speculative but interesting.

Additional comments

line 275 please add the word clade before B;
line 280, please add the word clade before C.

---

## Round 0.2 · accepted · Accept

Reviewer 2 found that his / her previous concerns were addressed in the revision. I was pleased to see that you added some additional molecular evidence in the revision, I still wonder about the likelihood that phylogenetic constraint explains the invasion pattern given that you have not presented any data on propagule pressure (introduction rates) of different genetic clades). However you did use non-definitive wording "suggesting the potential to become invasive is phylogenetically constrained." My personal preference for the abstract would be to use the phrase "may be" as in ""suggesting the potential to become invasive may be phylogenetically constrained.". But this would be my personal preference and if you prefer your current wording, that is OK with me.

Reviewer 2 ·

Basic reporting

The authors have clarified all the points mentioned in my previous review.

Experimental design

no comments

Validity of the findings

The authors have changed the manuscript in order to address the previous concerns